# Mechanical Behavior Characteristics and Energy Evolution Law of Coal Samples under the Influence of Loading Rate—A Case Study of Deep Mining in Wudong Coal Mine

Xingping Lai [1,2], Chong Jia [1,2,*], Feng Cui [1,2,3,*], Ganggui Feng [1,2], Mengqi Tian [1,2], Yifei Li [1,2] and Cheng Zong [1,2]

1   College of Energy Engineering, Xi'an University of Science and Technology, Xi'an 710054, China
2   Key Laboratory of Western Mines and Hazard Prevention of China Ministry of Education, Xi'an 710054, China
3   Key Laboratory of Coal Resources Exploration and Comprehensive Utilization of Ministry of Land and Resources, Xi'an 710021, China
*   Correspondence: jia654784376@163.com (C.J.); fengc@xust.edu.cn (F.C.); Tel.: +86-1552-561-9773 (C.J.)

**Abstract:** In order to clarify the mechanical properties and energy changes of coal samples under the influence of mining depth, a mechanical test analysis method to determine that the increase in mining depth increases the loading rate has been developed. Taking the Wudong Coal Mine as an example, a mechanical test analysis of coal samples is carried out. The results show that the surface deformation and failure of coal samples in the loading process presents four stages. That is, the evolution process of 'complete coal sample'–'partial failure-failure extension'–'overall instability'. The maximum temperature of a coal sample when it is destroyed shows an obvious nonlinear increasing trend with the increase in loading rate. With the increase in loading rate, the strength and elastic modulus of coal samples decrease gradually. The cumulative total energy and elastic energy of coal samples are linearly positively correlated with the loading rate. The research results provide ideas for rational control of mining intensity and determination of productivity in steeply inclined thick coal seams for deep mining.

**Keywords:** steeply inclined thick coal seam; loading rate; mechanical properties; energy evolution; elastic strain energy

## 1. Introduction

Steeply inclined thick coal seams are widely distributed in China, with large dip angles and complex geological structures. This makes the mining of steeply inclined thick coal seams a technical problem in mining that has long plagued the development of China's coal industry [1–9]. Therefore, in order to meet the safety and high-efficiency production of steeply inclined thick coal seam entering deep mining, it is urgent to carry out mechanical test research on coal samples affected by mining depth in order to better understand the energy accumulation and its evolution law in the process of mining stress increase under the influence of mining depth.

Loading rate has obvious influence on the mechanical properties and energy change characteristics of specimens. Many scholars have studied the strength, deformation, energy accumulation, and release characteristics of specimens under loading rate. Among them, Huang et al. established the rule that the strain increment of composite coal rock in elastic stage, plastic stage, and failure stage gradually increases with the increase in loading rate [10]. Zhao et al. determined that the crack initiation and propagation process of coal samples were recorded by high-speed and high-resolution digital camera, and also found that the effect of layer on crack morphology decreased with the increase in loading rate and crack length [11]. Lu et al.'s analysis shows that the higher the loading rate is, the more

fragile the post-peak behavior is, and the more serious the damage of coal sample is in the final loading stage [12]. Chen et al. found that the peak stress, strength softening modulus, elastic modulus, strain softening modulus, and post-peak modulus decrease partially with the increase in moisture content and loading rate [13]. Gong et al. found that the dynamic stress–strain curve of coal–rock combination has double peaks in the range of high loading rate [14]. Li et al. determine that the faster the loading rate is, the earlier the damage stress of the specimen appears and the faster the specimen is destroyed [15]. Wang et al., in considering the coal block structure of coal–rock composite structure, found that with the increase in loading rate, the three basic failure modes of composite structure are progressive shear failure, splitting failure, and structural failure [16]. Gao et al. found that when the loading rate is lower than the threshold, the uniaxial compressive strength and released elastic strain energy increase with the increase in loading rate. Otherwise, it may decrease with the loading rate [17]. Ma et al. found that with the increase in loading rate, the cumulative AE count decreases at first and then increases, and the total absorption energy and dissipative energy of coal–rock composite samples increase nonlinearly, while the released elastic strain energy increases at first and then decreases [18]. Lu et al. found that the peak stress of coal samples increased approximately logarithmically with the increase in loading rate, and it is considered that the increase in loading rate can enhance the intensity of disasters [19]. Xiao et al. found that with the increase in axial loading rate, the activity of AE signal increases, the threshold of strain level decreases with the active period and strong period of AE, and the AE signal with higher amplitude is produced in the intense period [20]. Xue et al. found that the maximum number of sound transmitting rings is small at a low loading rate. However, the maximum number of sound transmitting rings at a high loading rate increased significantly [21]. Cao et al. modeled the geomechanical and hydrological behaviors of broken coal under different stress loading conditions using a finite element model, and established the shear and tensile damage models of coal. [22]. Yin et al. observed the effect of loading rate on the mechanical behavior of roof-coal pillar structure [23]. Jiang et al. determined that when coal and rock are destroyed, the temperature changes abruptly, and the maximum infrared radiation temperature increases with the increase in loading rate [24]. Lai et al. carried out an analysis of the whole process of loading failure of the bearing rock sample with the help of scale transformation [25].

The mining depth of steeply inclined thick coal seam directly affects the loading rate of coal and rock mass during mining. Therefore, it is of great significance to study the influence of loading rate on the mechanical behavior of coal. The above-mentioned research mainly focuses on hard and brittle rocks, and there is relatively little research on weak coal and rock in steeply inclined thick coal seam mining. In view of this, this paper carried out uniaxial compression testing on a B3+6 coal seam in Wudong Coal Mine. Based on the variation characteristics of horizontal stress of coal under the influence of mining at different mining depths, the mechanical test analysis of increasing loading rate with the increase in mining depth is put forward and completed. The purpose is to determine the mechanical characteristics and energy change characteristics of coal samples in the process of increasing mining depth of steeply coal seam, and provide reference values for the safe mining of deep coal mines.

## 2. Engineering Background

Wudong Coal Mine, located in the eastern suburb of Urumqi, is a typical mining area with steeply 45~87° inclined thick coal seam. As shown in Figure 1, the mine location and mining interlayer layout of the working face shows that the occurrence dip angle of the coal and rock mass in the south area of Wudong Coal Mine is 87°, and the horizontal sublevel fully mechanized top coal caving mining method is adopted at the site. At present, in the south mining area of Wudong Coal Mine, B3+6 coal seam +425 level is mainly mined. Its sectional height is 25 m, coal cutting height is 3~5 m, and coal caving height is about 20 m.

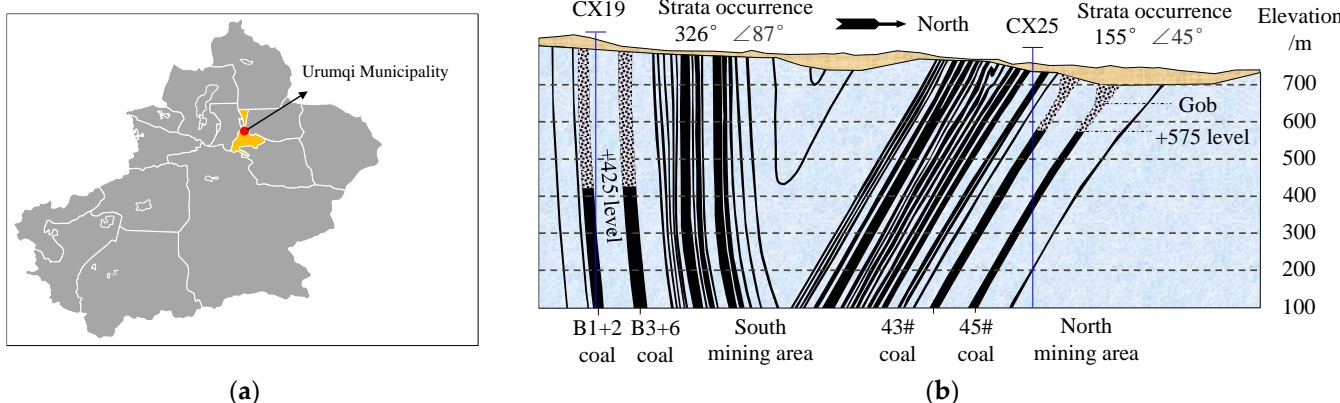

(**a**)　　　　　(**b**)

**Figure 1.** Mine location and mining interlayer layout of working face. (**a**) Mine location; (**b**) Layout of mining layers in working face.

　　Taking the B3+6 coal seam as an example, according to the team's previous in-situ stress detection results, the characteristics of horizontal stress variation under the influence of mining depth of steeply inclined thick coal seam are shown in Figure 2. The steeply inclined thick coal seam is mined to the depths of +475, +450, +425, and +400. The average horizontal stresses in the $x$ direction in front of the working face are 18.64, 19.18, 20.15, and 21.87 MPa, respectively. The average horizontal stress increases with the increase in mining depth, and its increase gradually increases, showing an obvious nonlinear increasing trend. The expression of its fitting curve is: $\sigma_x = 3.84 \times 10^{-4}\, e^{(h/43.32)} + 17.94$. Mining the steeply inclined thick coal seam to 475, +450, +425, and +400 depth, the maximum stress changes of the working face are 9.25, 10.18, 11.42, and 12.19 MPa respectively. With the increase in mining depth, the maximum horizontal stress of a single working face increases obviously. Therefore, this time, the coal is taken from a position which is not affected by the obvious disturbance caused by the mining of the working face—that is, the area 200 m deep in front of the B3+6 coal seam +425 horizontal working face—so as to analyze the mechanical characteristics and energy change characteristics of the coal sample under the influence of the increase in mining depth and loading rate. The sampling position and load direction are shown in Figure 3 below.

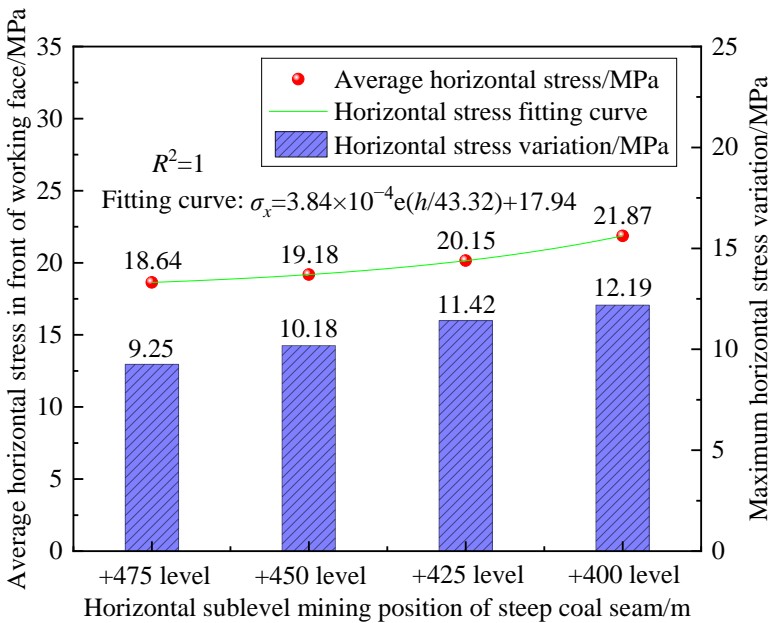

**Figure 2.** Characteristics of horizontal stress variation under the influence of mining depth.

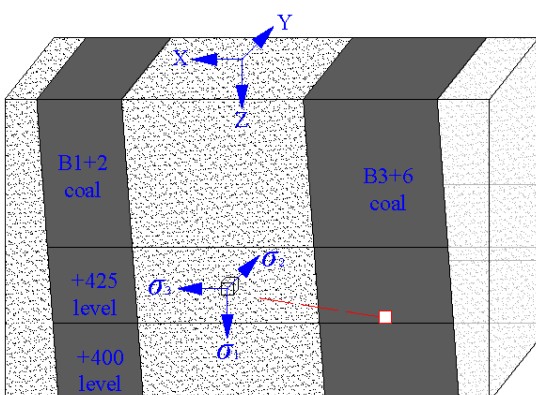

**Figure 3.** Schematic diagram of sampling position and load direction.

After the coal is taken from the site, it is made into a φ Φ50 × 100 mm cylinder standard coal samples which are then sealed to isolate the air and prevent weathering. As the coal samples can easily be weathered, plastic fresh-keeping film, plastic foam sponge, and plastic sealing tape were selected as the sealing materials to package the processed coal samples. The prepared and sealed briquette coal sample is shown in Figure 4. Sample preparation shall strictly control its sampling accuracy. The end faces shall be perpendicular to the axis of the sample, and the height error of coal and rock shall not exceed 0.3 mm.

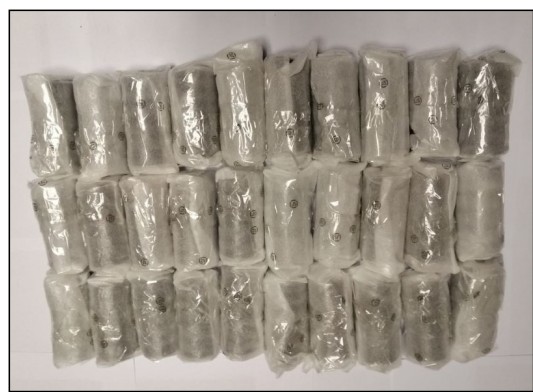

**Figure 4.** Prepared and sealed briquette coal samples.

## 3. Design of Mechanical Test Scheme of Coal Samples Affected by Loading Rate

According to the requirements of the national standard GB/T 25217.2-2010 "material testing machine is loaded at a rate of 0.5~1.0 MPa/s until the coal sample is destroyed". Firstly, the conventional uniaxial test of the coal samples with a loading rate of 0.50 MPa/s was carried out.

The axial displacement $L_0$ of natural coal samples was measured by the loading rate of 0.50 MPa/s specified in the national standard. The times from the start of the test to the failure were 5.0, 10.0, 15.0, and 20.0 min, respectively. Under equal displacement control, four different loading rates of $1/5L_0$, $1/10L_0$, $1/15L_0$, and $1/20L_0$ mm/min were carried out. The summary of the coal sample loading test scheme is shown in Table 1.

**Table 1.** Summary of coal sample loading test scheme.

| Number | Loading Rate | Number of Samples |
| --- | --- | --- |
| CD1, CD2, CD3 | 0.50 MPa/s | 3 |
| CD4, CD5, CD6 | $1/5L_0$ mm/min | 3 |
| CD7, CD8, CD9 | $1/10L_0$ mm/min | 3 |
| CD10, CD11, CD12 | $1/15L_0$ mm/min | 3 |
| CD13, CD14, CD15 | $1/20L_0$ mm/min | 3 |

Before the mechanical testing of coal samples at different loading rates, the wave velocity of coal samples should be measured at first. After coal samples with obvious differences in wave velocity are removed, the mechanical test research under the established scheme is carried out. The coal sample screening of RSM-SY5(T) nonmetallic acoustic testing equipment is shown in Figure 5. The acoustic wave test makes the microstructure of the screened coal sample smaller, such as primary cracks and structural planes. The wave velocities of the selected coal samples are all in the range of 1.4~1.5 km/s, and the wave velocities of the rejected coal samples are all outside this range.

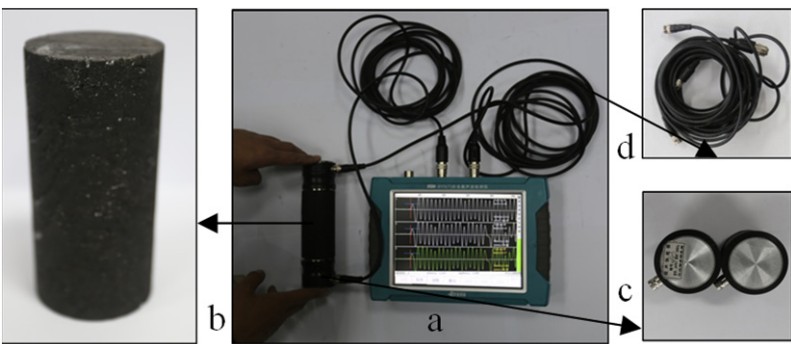

**Figure 5.** Coal sample screening of RSM-SY5(T) nonmetallic acoustic testing equipment. (**a**) RSM-SY5(T) nonmetallic acoustic detector host; (**b**) Sample to be tested; (**c**) Sandwich longitudinal wave planar transducer (50 KHz); (**d**) Connecting cable of planar transducer.

At present, acoustic emission technology—as a mature nondestructive testing method—has been widely used in many fields, such as coal and rock sample testing. Therefore, the loading system and acoustic emission monitoring system should be kept synchronized at a constant loading speed during the whole process of coal samples being directly damaged by stress and deformation. According to the analysis and inference of acoustic emission signals, the acoustic emission law of deformation and failure of coal samples at different loading rates can be grasped. The acoustic emission test system of MISTRAS produced by the American Physical Acoustics Company (PAC) is shown in Figure 6. The system is mainly composed of sensor, preamplifier, acquisition box, AEwin software display, etc. Based on the consideration of amplitude range, frequency distribution range, and rock acoustic emission signal, an R3 sensor is used in the test.

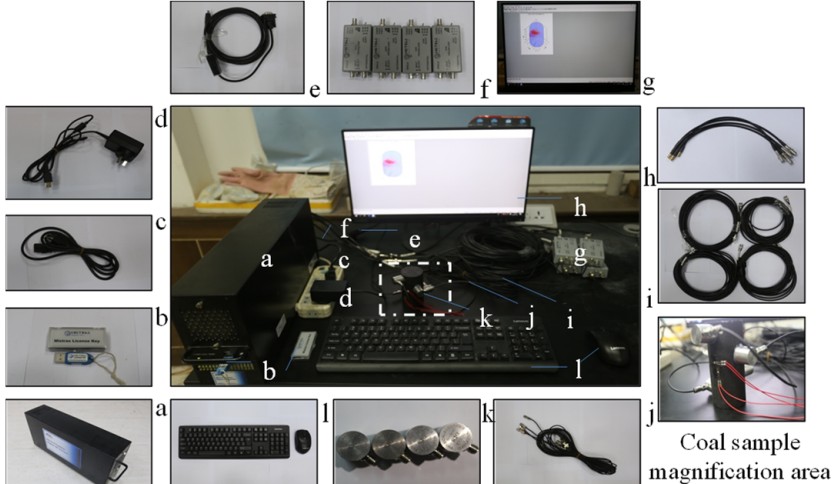

**Figure 6.** MISTRAS acoustic emission test system. (**a**) Mistras acoustic emission test host; (**b**) Combination lock; (**c**) Host power cord; (**d**) Power cord of the display; (**e**) Connecting line between host and display; (**f**) Amplifier; (**g**) Analysis software display; (**h**–**j**) Sensor connecting lines; (**k**) Highly sensitive $R_3$ acoustic emission sensor; (**l**) Software operating equipment.

## 4. Instantaneous Failure of Coal Sample under Loading and Its Temperature Variation Law

### 4.1. Instantaneous Failure Characteristics of Coal Samples during Loading

Under the loading rate of 0.50 MPa/s, through the real-time monitoring of a high-speed camera, the evolution process of coal sample surface deformation and failure is documented as shown in Figure 7.

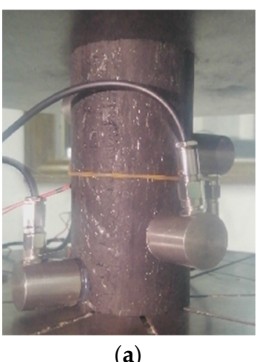 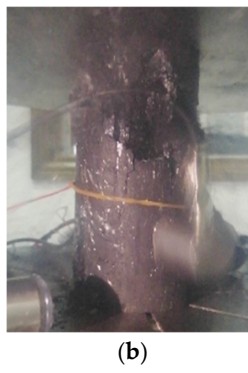 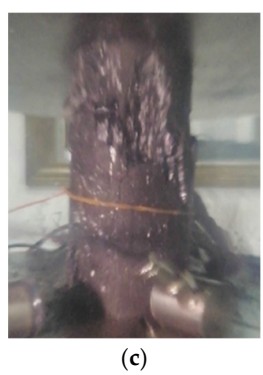 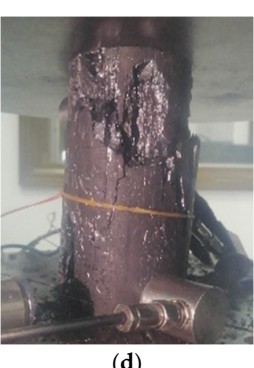

(**a**)  (**b**)  (**c**)  (**d**)

**Figure 7.** Evolution process of surface deformation and failure of coal samples. (**a**) Complete coal sample before the failure; (**b**) Coal sample in partial failure; (**c**) Failure extension; (**d**) Overall instability.

It can be seen from Figure 7 that the surface deformation and failure of coal samples show four obvious processes "complete coal samples-partial failure-failure extension-overall instability". Among them, there is no obvious difference in the surface deformation characteristics between the complete coal sample before the failure (Figure 7a) and the coal sample before the initial compression test. In the continuous loading process of the testing machine, after reaching the critical position of failure, the coal sample is under partial failure (Figure 7b), then failure extension (Figure 7c), and finally overall instability (Figure 7d).

### 4.2. Characteristics of Surface Temperature Change of Coal Samples during Loading

Thermal monitoring can realize the temperature capture in the process of sample loading [26]. Therefore, the thermal infrared camera is used for real-time monitoring, and the evolution characteristics of surface temperature of coal sample failure are shown in Figure 8.

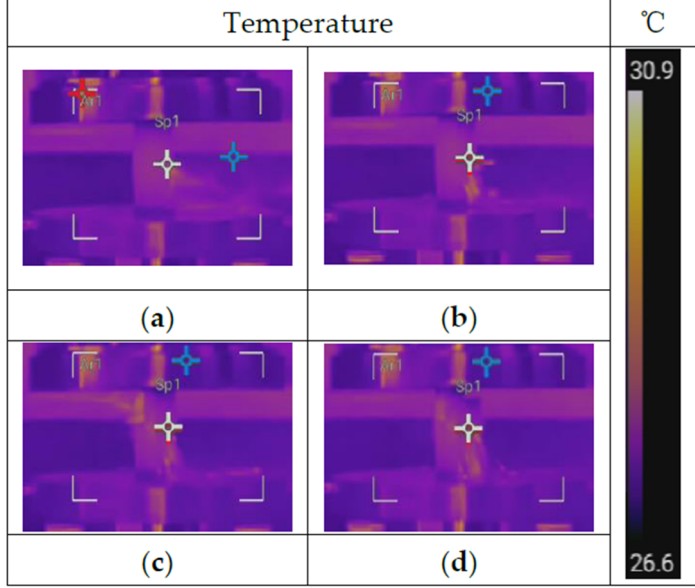

**Figure 8.** Evolution characteristics of surface temperature of coal sample failure. (**a**) Local rupture; (**b**) Partial destruction; (**c**) Damage extension; (**d**) Overall instability.

According to the evolution characteristics of surface temperature of coal samples from loading to failure, the surface deformation and failure of coal samples showed four obvious processes: "local fracture-partial failure-failure extension-overall instability". The middle part of the squeezed coal sample is deformed greatly, and the local fracture of the coal sample occurs when it reaches the critical position of failure (Figure 8a), and the rock block in the middle part of the coal sample popped out instantly. The continuous loading of the press makes the coal sample partially destroyed (Figure 8b), and the block at the lower part of the coal sample broke in a large area. Subsequently, the failure extended (Figure 8c) so that the upper block of the coal sample was broken, and after the upper and lower broken blocks were connected, the overall instability of the coal sample is formed (Figure 8d).

Before testing the different loading rates of displacement control, the average displacement $L_0$ of 0.50 MPa/s coal sample under conventional loading (That is 1.17 mm) was extracted. The different loading rates of coal samples from low to high were determined to be $9.75 \times 10^{-4}$ mm/s, $1.30 \times 10^{-3}$ mm/s, $1.95 \times 10^{-3}$ mm/s, and $3.90 \times 10^{-3}$ mm/s.

According to the surface temperature monitoring of coal samples at different loading rates, the average temperature of a single group of coal samples from loading to failure is taken as an example. The change trend of surface temperature of coal samples affected by different loading rates is shown in Figure 9. It can be seen from Figure 9 that under the different loading rates of $9.75 \times 10^{-4}$ mm/s, $1.30 \times 10^{-3}$ mm/s, $1.95 \times 10^{-3}$ mm/s, and $3.90 \times 10^{-3}$ mm/s. The maximum temperatures at the time of coal sample failure were 29.1 °C, 30.2 °C, 31.4 °C, and 32.3 °C, and the maximum temperature differences per unit of time were 10.1 °C, 10.8 °C, 11.3 °C, and 11.7 °C, respectively. The maximum temperature and the maximum temperature difference per unit time when the coal sample is destroyed show an obvious nonlinear increasing trend with the increase in loading rate. Taking the maximum temperature difference per unit time as an example, the nonlinear expression under the influence of loading rate is as follows $\Delta T = 11.697 - 7.188 \times 0.211^V$, with the increase in loading rate, the maximum temperature difference per unit time decreases gradually.

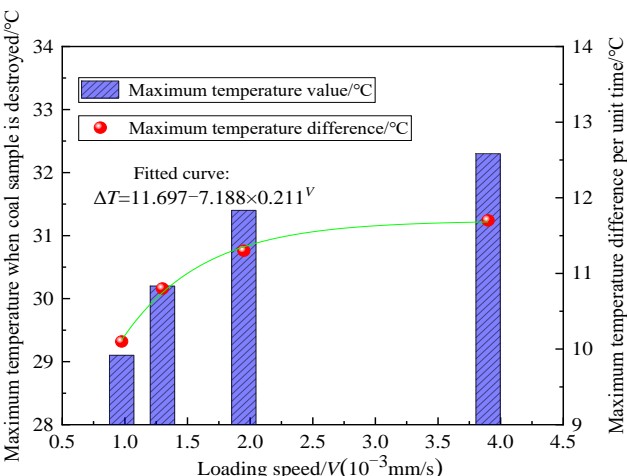

**Figure 9.** Change trend of surface temperature of coal samples affected by different loading rates.

## 5. Mechanical Test Analysis of Coal Samples

### 5.1. Mechanical Properties of Coal Samples under the Influence of Loading Rate

The process of deformation and failure of coal under compression usually goes through four stages: energy input, accumulation, dissipation, and release. A scholar has defined the cumulative dissipated energy density of the experiment [27]. It is assumed that the temperature is constant during the test, and there is no heat exchange between the physical process and the outside world. According to the law of conservation of energy, the following relation exists [28–30]:

$$U^d = U - U^e \tag{1}$$

where $U$ is the total energy absorbed by the coal sample, $U^d$ is the energy dissipated by the coal sample, and $U^e$ is the elastic strain energy accumulated inside the coal sample.

Total energy absorbed by coal sample is

$$U = \int \sigma_1 d\varepsilon_1 \tag{2}$$

where $\sigma_1$ is the main stress, and $\varepsilon_1$ is the main strain.

The elastic energy stored in the coal sample is

$$U^e = 0.5\sigma_1\varepsilon_1^e \approx 0.5\sigma_1^2/E_0 \tag{3}$$

where $\varepsilon_1^e$ is the elastic value, and $E_0$ is the initial elastic modulus.

According to the coal sample failure characteristics of conventional loading tests and the calculation results of Equations (1)–(3), the coal sample failure and strain energy change characteristics affected by loading rate are drawn as shown in Figure 10.

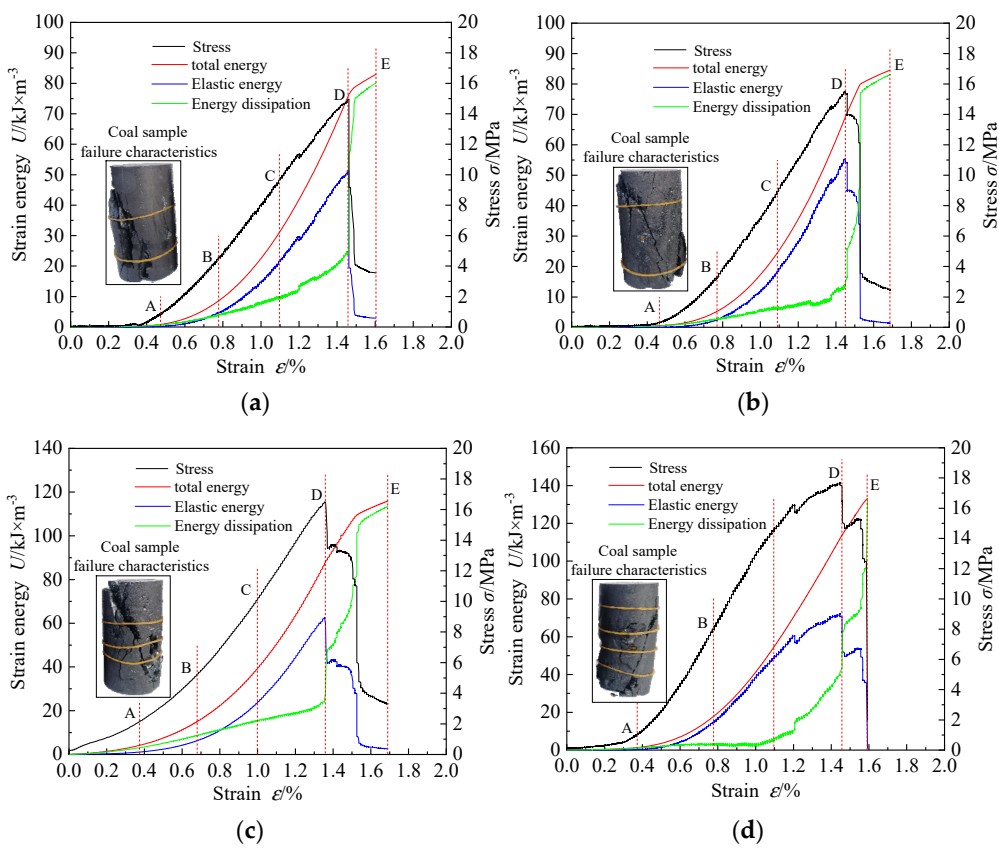

**Figure 10.** Coal sample failure affected by loading rate and its strain energy change characteristics. (**a**) Energy evolution curve of $9.75 \times 10^{-4}$ mm/s coal sample. (**b**) Energy evolution curve of $1.30 \times 10^{-3}$ mm/s coal sample. (**c**) Energy evolution curve of $1.95 \times 10^{-3}$ mm/s coal sample. (**d**) Energy evolution curve of $3.90 \times 10^{-3}$ mm/s coal sample.

As can be seen from Figure 10, the total strain energy $U$ gradually increases with the increase in strain, and the increasing amplitude shows the trend of "increasing-stabilizing-slowing down". With the increase in strain, the elastic energy shows a trend of "increasing first, reaching the peak and then decreasing". When the stress reaches the peak value, the elastic energy also reaches the peak value, and with the destruction of coal samples, the elastic energy with only residual strength drops to near zero. The dissipation energy increases gradually with the increase in strain, with the most obvious increase in the failure stage being found after the peak of coal samples. Due to the difference of joints and fissures

in coal samples, the anisotropy of coal samples is obvious. Therefore, under the influence of different loading rates, the shape of its stress–strain curve has obvious interval differences.

In the stage of region 0~A, the stress loading in the elastic range of the curve is quite different due to the influence of primary cracks. In the stage A~B, the curve slope—namely, the elastic modulus—is obviously higher at the loading rate of $3.90 \times 10^{-3}$ mm/s. During the period from B to D, with the increasing of axial stress, the plastic deformation is obvious, and its stress–strain curve gradually evolves into a concave shape. Due to the difference of loading rate, the stress loading amount and strain distribution range in the plastic range are obviously different. In the post-peak stage, the residual strength of coal samples is small when the loading rate is $3.90 \times 10^{-3}$ mm/s, while the softening of post-peak stress is more obvious when the loading rate is small, and the retained residual strain is larger. Under different loading rates controlled by displacement, the failure form of coal samples is mainly shear failure. Moreover, the failure angles of coal samples at different loading rates are all in the range of 66~75, and the failure angles show a weak increasing trend with the increase in loading rate. The damage angle of coal samples increases slightly with the increase in loading rate. The degree of coal breakage increases with the increase in loading rate, and the faster the loading rate, the more broken the coal sample is.

The variation law of peak strength and elastic modulus of coal samples affected by loading rate is shown in Figure 11. From the relationship between peak strength of coal samples and loading rate in Figure 11a, the average strengths of coal samples at different loading rates of $9.75 \times 10^{-4}$, $1.30 \times 10^{-3}$, $1.95 \times 10^{-3}$, and $3.90 \times 10^{-3}$ mm/s are 14.73, 15.66, 16.36, and 16.99 respectively. According to the fitting results of the average strength of each group of coal samples, it can be seen that the peak strength of coal samples has an exponential function relationship with the fitting degree of 0.99546, and the fitting curve expression is $\sigma = -8.75358 \times \mathrm{e}^{-V/0.71563} + 17.00396$. The strength of coal samples increases with the increase in loading rate, and the strength increase decreases with the increase in loading rate. According to the relationship between elastic modulus of coal sample and loading rate in Figure 11b. The average elastic moduli of coal samples at different loading rates of $9.75 \times 10^{-4}$, $1.30 \times 10^{-3}$, $1.95 \times 10^{-3}$, and $3.90 \times 10^{-3}$ mm/s are 1.76, 1.88, 1.99, and 2.20 GPa respectively. The elastic modulus of coal samples and the loading rate show an exponential function relationship with a fitting degree of 0.99503, and the fitting curve is expressed as: $E = -0.93503 \times \mathrm{e}^{-V/1.63539} + 2.28433$. Therefore, in a certain range, the strength and elastic modulus of coal samples increase with the increase in loading rate, and the increase rate gradually decreases.

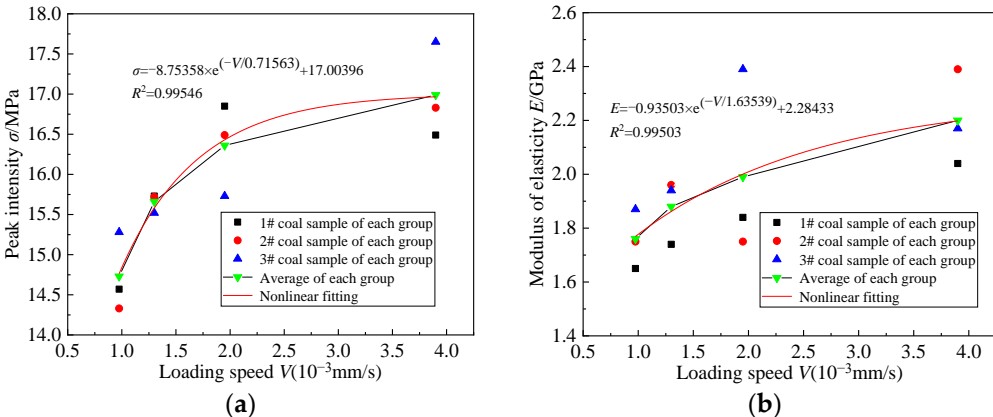

**Figure 11.** Changes of peak strength and elastic modulus of coal samples affected by loading rate. (**a**) Relationship between peak strength of coal sample and loading rate. (**b**) Relationship between elastic modulus of coal sample and loading rate.

## 5.2. *Effect of Loading Rate on Strain Energy and Acoustic Emission Energy of Coal Samples*

The deformation and failure process of coal samples during loading is also a process of internal micro-defects constantly evolving, which is often accompanied by energy dissi-

pation, and the initial stage of its stress–strain curve has low dissipation energy. Therefore, elastic strain energy rate $\alpha = U^e/U$ and dissipation energy rate $\beta = U^d/U$ are defined to better represent the energy dissipation during loading. Taking normalized stress-real-time stress $\sigma_i$/peak stress $\sigma_p$ as the abscissa and elastic strain energy rate $\alpha$ and dissipation energy rate $\beta$ as the ordinate. The energy rate variation curves at different loading rates are drawn as shown in Figure 12, and the energy dissipation rate variation law at quasi-static loading rate is analyzed.

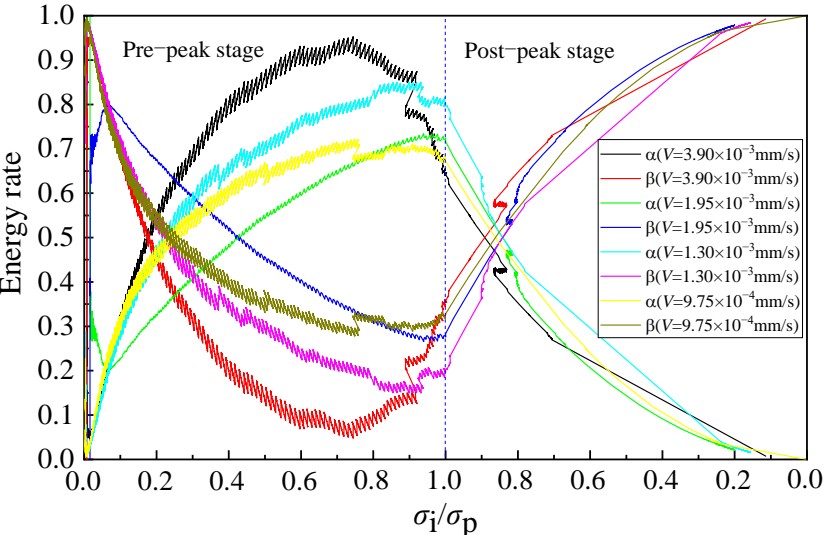

**Figure 12.** Energy rate changes at different loading rates.

It can be seen from Figure 12 that the elastic strain energy rate $\alpha$ and dissipation energy rate $\beta$ at different loading rates have similar trends. Before the peak stress, with the increase in stress level, the elastic strain energy rate $\alpha$ first drops sharply, then the rate gradually decreases, and finally decreases slowly. Dissipative energy rate $\beta$ increases suddenly at first, then the rate decreases gradually, and finally increases slowly. Because the crack closure leads to the increase in the proportion of dissipated energy, $\alpha$ gradually decreases and $\beta$ gradually increases in the initial compaction stage. In the middle stage, the continuous energy absorption of coal and rock units and the increase in elastic deformation of coal and rock lead to the gradual increase in $\alpha$ and the gradual decrease in $\beta$. At the same time, cracking initiates, and the increasing rate of energy dissipation decreases gradually. In the pre-peak stage, coal and rock are mostly in the stage of rapid crack propagation. At this stage, the evolution of various microscopic defects of coal and rock may be amplified to form macroscopic cracks, which will lead to the instability of coal and rock. At this stage, both $\alpha$ and $\beta$ showed a fluctuating trend, and the overall change form was that $\alpha$ gradually decreased and $\beta$ gradually increased. As the loading continues, that is, in the post-peak stage, the elastic strain energy begins to release, and the crack propagation and the friction between the crack surfaces lead to the continuous increase in dissipation energy so that $\alpha$ decreases and $\beta$ increases continuously.

Dissipative energy is closely related to the strength of the sample, and its sudden change represents the aggravation of the damage inside the sample [31]. The higher the loading rate at the peak stage is, the higher the elastic strain energy rate $\alpha$ of coal samples is generally, and its dissipation energy rate is relatively small in the range of $\sigma_i/\sigma_p$ from 0 to 0.9. However, the dissipation energy rate in the range of $\sigma_i/\sigma_p$ from 0.9 to 1.0 is relatively large, and the dissipation energy rate is high before reaching the peak stress ($\sigma_i/\sigma_p = 1$). With the deep mining of steep thick coal seams, the greater the loading rate of coal sample, the smaller the internal damage degree of coal sample must be before failure, and the higher the accumulated strain energy. The internal damage of the coal sample is obviously

aggravated before it is destroyed, which makes the accumulated strain energy release intensively, and it is easier to induce dynamic instability.

The energy parameters at different loading rates are statistically summarized, and the average value of the energy parameters at the stress peak at different loading rates is taken as an example. The variation curves of strain energy at different loading rates are shown in Figure 13. The total strain energy and elastic energy of $9.75 \times 10^{-4}$, $1.30 \times 10^{-3}$, $1.95 \times 10^{-3}$, and $3.90 \times 10^{-3}$ mm/s are 77.02, 79.53, 81.47, and 89.35 kJ/m³, respectively. The total energy and elastic energy of coal samples increase with the increase in loading rate, and there is an obvious linear relationship between them. The fitting degrees are 0.9915 and 0.94113, respectively, and the expressions are: $U = 4.04813V + 73.61974$, $U^d = 3.73586V + 49.46154$. The difference of dissipation energy at different loading rates is small, and the dynamic change trend of dissipation energy and loading rate is less.

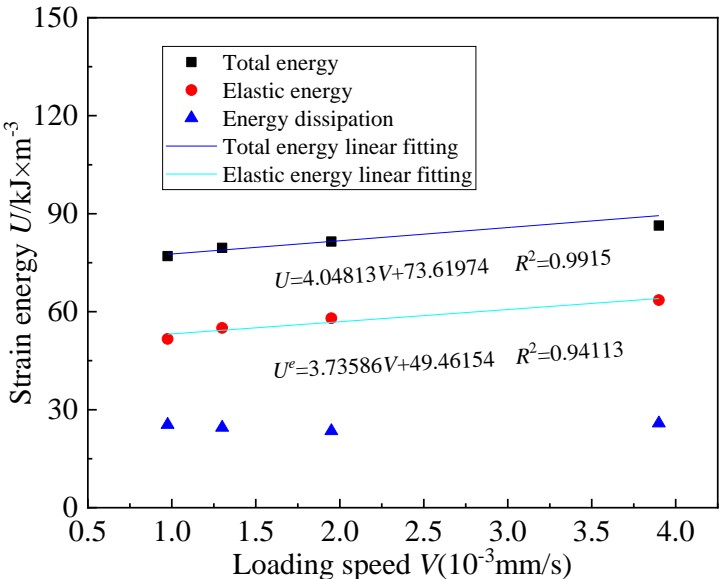

**Figure 13.** Variation curves of strain energy at different loading rates.

The acoustic emission events affected by loading rate and their energy variation characteristics are shown in Figure 14. The whole loading process of coal samples can be divided into three different areas: "microcrack incubation period", "concentrated fissure development period", "late repture". Due to the time point when the number of acoustic emission events rises sharply, the initial step-by-step increase in acoustic emission energy occurs, and the obvious stress drop is caused by coal sample failure. Among the three areas, the cumulative energy of acoustic emission in the "microcrack incubation period" increased slightly, while the "concentrated fissure development period" mainly occurred in the partial elastic-plastic stage (BC) and the whole stage of rapid fissure expansion (CD) during the loading process of coal samples. During this period, the cumulative energy of acoustic emission increased greatly, and the energy of instantaneous excitation greatly increased, and then the coal samples were destroyed.

With the increase in loading rate, the time from loading to failure of coal samples decreases, and the acoustic emission energy measured in the concentrated development period of cracks increases sharply. Under the loading rates of $9.75 \times 10^{-4}$, $1.30 \times 10^{-3}$, $1.95 \times 10^{-3}$, and $3.90 \times 10^{-3}$ mm/s, the acoustic emission energy per unit time is 71.33, 98.76, 267.51, and 516.13 mV·us/s, respectively. The cumulative energy of acoustic emissions of coal samples from loading to failure are $6.29 \times 10^4$, $6.68 \times 10^4$, $7.83 \times 10^4$, and $8.65 \times 10^4$ mV·us, respectively. The acoustic emission energy of coal samples per unit time increases with the increase in loading rate, and the increase rate gradually increases. At the same time, the cumulative energy of acoustic emission of coal samples from loading to failure gradually increases with the increase in loading rate.

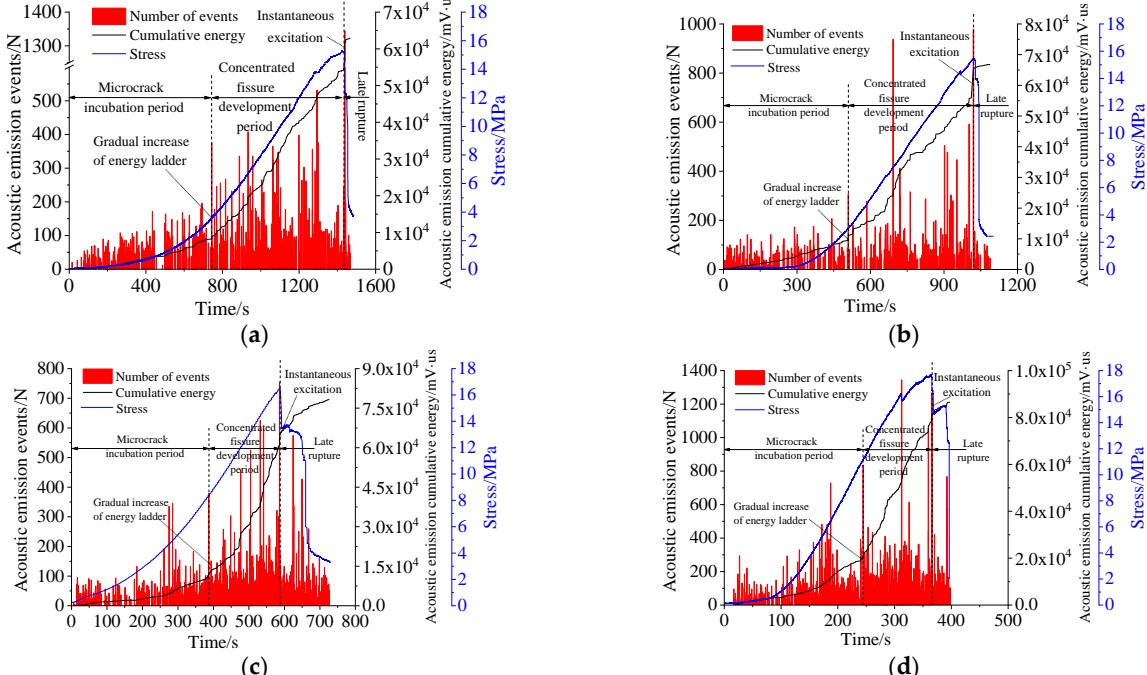

**Figure 14.** Acoustic emission events affected by loading rate and their energy variation characteristics. (**a**) $9.75 \times 10^{-4}$ mm/s acoustic emission energy variation curve. (**b**) $1.30 \times 10^{-3}$ mm/s acoustic emission energy curve. (**c**) $3.90 \times 10^{-3}$ mm/s acoustic emission energy curve. (**d**) $3.90 \times 10^{-3}$ mm/s acoustic emission energy curve.

## 6. Discussion

With the increase in mining depth, the loading rate under the influence of coal mining increases. Therefore, through the mechanical test results of coal samples with different loading rates, the mechanical properties and energy changes of coal samples caused by the increase in coal seam mining depth are comprehensively analyzed. See Table 2 for mechanical properties of coal samples as affected by loading rate and statistics of their energy change. The "maximum temperature difference per unit time" is "the maximum temperature change of the test surface per unit time"; "unit time" is "time change per second".

**Table 2.** Mechanical properties of coal samples affected by loading rate and statistics of their energy change.

| Coal Loading Rate (mm/s) | $9.75 \times 10^{-4}$ | $1.30 \times 10^{-3}$ | $1.95 \times 10^{-3}$ | $3.90 \times 10^{-3}$ |
|---|---|---|---|---|
| Average peak strength of coal sample failure/MPa | 14.73 | 15.66 | 16.36 | 16.99 |
| Strain of coal sample when it produces stress drop/% | 1.46 | 1.52 | 1.51 | 1.60 |
| Average elastic modulus of coal sample failure/GPa | 1.76 | 1.88 | 1.99 | 2.20 |
| The highest temperature when coal sample is destroyed/°C | 29.1 | 30.2 | 31.4 | 32.3 |
| Maximum temperature difference per unit time of coal sample/°C | 10.1 | 10.8 | 11.3 | 11.7 |
| Total input energy at peak position of coal sample/kJ $\times$ m$^3$ | 77.02 | 79.53 | 81.47 | 89.35 |
| Elastic energy at peak position of coal sample/kJ $\times$ m$^3$ | 51.64 | 55.01 | 58.02 | 63.53 |
| Dissipative energy at the peak of coal sample/kJ $\times$ m$^3$ | 25.38 | 24.52 | 23.45 | 23.15 |
| Accumulated energy of acoustic emission in the whole loading process $10^4$/mV·us | 6.29 | 6.68 | 7.83 | 8.65 |
| Acoustic emission energy per unit time in concentrated fracture development period/mV·us/s | 71.33 | 98.76 | 267.51 | 516.13 |

From the statistical results in Table 2, it can be seen that the greater the loading rate of coal samples, the higher the stress and the greater the strain when coal samples are damaged. When the loading rate is $3.90 \times 10^3$ mm/s, its rate reaches 16.99 MPa. These kinds of characteristics create conditions for the energy accumulation of coal samples, which makes the maximum temperature and maximum temperature difference increase obviously. At the loading rate of $3.90 \times 10^3$ mm/s, the total input energy and elastic energy of coal samples loaded to the peak position are 89.35 kJ $\times$ m$^3$ and 63.53 kJ $\times$ m$^3$, respectively. The higher the loading rate, the higher the stress and strain energy of the coal sample when it is loaded to the peak position, and the higher the energy released is when the coal sample is destroyed. When the loading rate is $3.90 \times 10^3$ mm/s, the maximum cumulative energy of acoustic emission in the whole loading process of coal samples reaches $8.65 \times 10^4$ mV·us.

Based on the variation characteristics of horizontal stress of coal under the influence of mining at different mining depths, the mechanical test analysis of increasing loading rate with the increase in mining depth is put forward and completed. It is concluded that the average compressive strength of coal samples increases nonlinearly with the increase in mining depth. The greater the loading rate of coal sample, the higher the stress of the coal sample when it is destroyed, and the greater the strain of failure. The greater the accumulated total energy and elastic energy, the higher the acoustic emission energy released by coal sample destruction. The nonlinear increase in coal sample failure strength and the gradual increase in accumulated strain energy in the process of increasing loading rate with increasing mining depth are studied. It provides a basis for further strengthening the support strength and continuous optimization of mining parameters in deep mining engineering.

The mining conditions of steeply inclined thick coal seams are complex, and the dynamic disasters—such as rock burst—show obvious shallow buried depth characteristics. In the process of horizontal sublevel mining, the mining depth of the coal seam gradually increases. In the process of increasing the loading rate of coal samples, the temperature and energy of coal samples when destroyed obviously increase, which results in an upward trend for dynamic disasters—such as gas outburst and rock burst.

Microseismic monitoring can realize the real-time collection of energy data in the process of mine production [32]. During the non-impact event and when the mining speed is relatively consistent, the on-site microseismic monitoring results within 30 days of continuous advancing of the working face will be analyzed. Because the mine is currently mined to the +425 level, only microseisms at the +475, +450, and +425 levels are counted. The average microseismic energy in the mining process of the +475, +450, and +425 levels is $2.22 \times 10^4$ J, $3.06 \times 10^4$ J, and $3.93 \times 10^4$ J respectively, and the average microseismic frequencies are 22, 25, and 32 respectively. The field microseismic energy and frequency increase with the increase in mining depth, and its amplitude gradually increases, which effectively verifies the reliability of the energy variation law obtained from the test results.

In this paper, the change of stress and energy release characteristics under the influence of mining in coal seams is mainly studied, and the loading rate of coal seams at different mining depths is influenced. The analysis of mechanical characteristics and energy change characteristics of coal samples provides ideas for the rational regulation of advancing speed of coal seams to deep mining and the determination of productivity.

## 7. Conclusions

(1) With the increase in mining depth, the maximum horizontal stress variation of a single mine in its working face obviously increases, that is, the increase in mining depth makes the loading rate under the influence of coal mining increase. Therefore, this paper innovatively puts forward a new method to analyze the mechanical characteristics and energy changes of coal samples affected by the increase in mining depth through mechanical tests of coal samples with different loading rates.

(2)  During the loading process, the deformation and failure of coal sample surface showed four obvious processes—namely, the evolution process of "complete coal sample-partial failure-failure extension-overall instability". Under the different loading rates of $9.75 \times 10^{-4}$, $1.30 \times 10^{-3}$, $1.95 \times 10^{-3}$, and $3.90 \times 10^{-3}$ mm/s, the maximum temperature and the maximum temperature difference per unit time of coal sample failure show an obvious nonlinear increasing trend with the increasing in loading rate.

(3)  The strength and elastic modulus of coal samples are exponential functions with a high fitting degree with the loading rate, and the increase rate decreases gradually with the increase in loading rate. The cumulative total energy and elastic energy of coal samples are linearly positively correlated with the loading rate. The higher the loading rate in the front stage, the higher the dissipation energy rate before reaching the peak stress, and the greater the cumulative energy of acoustic emission released from loading to failure of coal samples.

(4)  The higher the loading rate of a coal sample, the higher the stress and strain of that coal sample when it is destroyed, which makes the maximum temperature and maximum temperature difference of the coal sample when it is destroyed obviously increase. The greater the total energy input and elastic energy, the higher the acoustic emission energy released when the coal sample is destroyed. With the deep mining of coal seams, the cumulative strain energy of the coal sample is higher. The more serious the internal damage is before the coal sample is destroyed, the more easily concentrated release the accumulated strain energy induces dynamic instability.

**Author Contributions:** X.L. conceived, designed, and analyzed the test results; C.J. performed the experiments and wrote the manuscript; F.C. handled project administration; G.F. and M.T. prepared coal samples; Y.L. and C.Z. participated in the processing of some data. All authors have made equal contributions to the revision of the paper. All authors have read and agreed to the published version of the manuscript.

**Funding:** This work was sponsored by the National Natural Science Foundation of China (no. 51874231); Shaanxi Natural Science Fundamental Research Program Enterprise United Fund (2019JLZ-04); Shaanxi Province Innovation Ability Support Plan Project (2020KJXX-006).

**Institutional Review Board Statement:** No applicable.

**Informed Consent Statement:** No applicable.

**Data Availability Statement:** The data used to support the findings of this study are available from the corresponding author upon request.

**Acknowledgments:** The author is sincerely thankful for the equipment and site support provided by Key Laboratory of Western Mines and Hazard Prevention of China Ministry of Education and Key Laboratory of Coal Resource Exploration and Comprehensive Utilization, and also for the funding support provided by those mentioned above.

**Conflicts of Interest:** The authors declare no conflict of interest.

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
