# Peer review of "Mechanical Behavior Characteristics and Energy Evolution Law of Coal Samples under the Influence of Loading Rate—A Case Study of Deep Mining in Wudong Coal Mine"

_minerals, doi:10.3390/min12081032_

Round 1

Reviewer 1 Report

In this manuscript, aiming at the mechanical characteristics and energy changes of coal samples under the influence of mining depth in steep thick coal seam, taking steep thick coal seam in Wudong Coal Mine as the background, the mechanical test of coal samples was conducted to analyze the stress changes affected by mining depth. This paper innovatively puts forward a mechanical test analysis method to study the change of mechanical properties and energy of coal samples with the increase of mining depth through mechanical tests of coal samples at different loading rates. Due to the following minor shortcomings, this reviewer suggest to be published after minor revised.

Page 5, line 167-168: RSM-SY5(T) nonmetallic sound wave detection equipment is used here to screen coal samples through the difference of wave velocity. However, what are the common characteristics of the wave velocity of the coal samples after this screening, and what are the differences in the wave velocity? The author needs further explanation.

Page 9, line 280-284: It is mentioned in this paper that under different loading rates controlled by displacement, the failure form of coal samples is mainly shear failure; And the failure angle of coal samples at different loading rates increases weakly with the increase of loading rate. However, under the influence of loading rate, there are not only differences in the damage angle of coal samples, but also obvious differences in the crushing degree. I hope the author can supplement this.

Page 13: In the chapter of "Discussion" in this paper, the author mainly makes a qualitative analysis on the mechanical properties and energy release characteristics of coal samples under the influence of loading rate, but it needs to be described quantitatively to reflect the data results of statistical analysis in Table 2.

In addition, there are still several minor sentence translation problems in the article, which should be checked and improved by the author. Part of the abstract and conclusion is too lengthy and needs to be appropriately simplified.

Author Response

Thanks to the valuable opinions provided by the experts, it is of great help to improve the article again. See the annex for the detailed reply.

Reviewer 2 Report

The laboratory tests on coal are widely carry put over the world. However there are still some points for discussion.

In this case authors show the influence of strain rate for coal behaviuor under loading. The topic is not worn out, so it might be interesting for readers but the manuscript needs some improvements. First of all some statements, comments and conclusions must be improved.

1. The authors use the term "loading rate" but they change the strain rate - IT MUST BE CHANGED

2. I guess that "steep 45-87" means 45-87 degreees. Lack of unit. (page 3)

3. What do you mean with: "previous simulation experiments"? When was it? (page3)

4. I cannot believe that the acccuracy of sample making was 0.1 mm, if they were sealed in plastic tape and sponge... (page 4)

5. I don't understand the time control if you want to get the peak stress having a set strain rate (Page 5)

6. What does it mean - the coal samples with obvious difference in wave velocity were removed"? - removed from where? How many were taken into account then? (page 5)

7. The description of four coal damage stages should the same in text (page 6)

8. "the elastic energy with only residual strength" - can you say about any elastic energy having broken coal pieces? (page 8)

9. That what intrigued me is how did you find the elastic energy and dissipated energy? Even using eq. 1-3 for linear section of sigma-epsilon elastic energy cannot drop down or fluctuated (fig. 10). You must explain it better or calculate it again.

10. You missed the comments aboputelastic and dissipated energy (change alpha and beta coefficients) in comments in Par. 5.2. (page 10)

11. Adjective "volatile" is improper in this context (page 11)

12. I cannot agree with the comments on page 12. the increase of cumulative energy of acoustic energy is not visible. And there is not any consequence in the results. Are you sure you don't mix the pictures?

13. Acoustic energy unit is "mv us/s". What is this? Not mV ms/s?

14. What is: "Max. temperature difference per unit time"? What "unit time"? What "difference" - explain (page 13)

15. Just before conclusion paragraph and in conclusions you mix "steeply inclined thick coal" and "coal samples" . You cannot start your comments with the rock mass and then say that "with deep mining... the degree of internal damage coal sample is small". The coal in the rock mass is a rock mass. Samples consider laboratory test. and why you have involved there "steeply inclined coal"? You didn't consider anisotropy. Conclusions and comments MUST BE RE-WRITTEN.

And English language check is required.

You can see how to describe the similar experiment properly and how accumulated dissipated energy density were defined:

Bai J., Dou L-M, Małkowski P., Li J., Zhou K. Chai Y. Mechanical Properties and Damage Behavior of Rock-Coal-Rock Combined Samples under Coupled Static and Dynamic Loads. Geofluids, Vol. 2021, Article ID 3181697, https://doi.org/10.1155/2021/3181697.

Author Response

(The authors gave the same response as above.)

Reviewer 3 Report

Please read the attached file

Author Response

(The authors gave the same response as above.)

Round 2

Reviewer 3 Report

Find the attached file

Author Response

Thanks to the experts for their valuable opinions. The author has provided a point-by-point response to the reviewers' comments. The details of the response are shown in the annex.
